# Alternate oscillations of Martian hydrogen and oxygen upper atmospheres during a major dust storm

Kei Masunaga [1] ✉, Naoki Terada[2], Nao Yoshida[2], Yuki Nakamura[2,3], Takeshi Kuroda [2,4], Kazuo Yoshioka [5,6], Yudai Suzuki [6], Hiromu Nakagawa[2], Tomoki Kimura[7], Fuminori Tsuchiya[8], Go Murakami[1], Atsushi Yamazaki [1], Tomohiro Usui [1] & Ichiro Yoshikawa[5,6]

Dust storms on Mars play a role in transporting water from its lower to upper atmosphere, seasonally enhancing hydrogen escape. However, it remains unclear how water is diurnally transported during a dust storm and how its elements, hydrogen and oxygen, are subsequently influenced in the upper atmosphere. Here, we use multi-spacecraft and space telescope observations obtained during a major dust storm in Mars Year 33 to show that hydrogen abundance in the upper atmosphere gradually increases because of water supply above an altitude of 60 km, while oxygen abundance temporarily decreases via water ice absorption, catalytic loss, or downward transportation. Additionally, atmospheric waves modulate dust and water transportations, causing alternate oscillations of hydrogen and oxygen abundances in the upper atmosphere. If dust- and wave-driven couplings of the Martian lower and upper atmospheres are common in dust storms, with increasing escape of hydrogen, oxygen will less efficiently escape from the upper atmosphere, leading to a more oxidized atmosphere. These findings provide insights regarding Mars' water loss history and its redox state, which are crucial for understanding the Martian habitable environment.

It is believed that, over billions of years, Mars has lost water through atmospheric escape. A classic Martian water loss mechanism is that $H_2$ molecules that are chemically produced from water vapor in the lower atmosphere diffuse into the upper atmosphere, and their subsequent dissociation results in hydrogen escape[1]. As the time scale of $H_2$ diffusion is 1200 years[2], it was believed that Mars' hydrogen escape rates have been steady within a shorter time scale. However, many observations have demonstrated that hydrogen escape rates from the Martian upper atmosphere vary seasonally due to seasonal change in vertical water vapor distribution[3–7]. Additionally, recent observations revealed that major dust storms, which expand regionally or globally on Mars, directly transfer water vapor from the lower to the upper atmosphere and rapidly change the vertical water distribution, increasing hydrogen escape[5–13]. Because major regional dust storms occur seasonally and global dust storms occasionally on Mars, monitoring Mars' hydrogen upper atmosphere and dust storms is crucial for understanding its water escape mechanisms. Recent studies have reported that the amount of oxygen atoms likely decreases during

[1]Institute of Space and Astronautical Science, Japan Aerospace Exploration Agency, Sagamihara, Japan. [2]Department of Geophysics, Graduate School of Science, Tohoku University, Sendai, Japan. [3]LATMOS, Sorbonne Université, Paris, France. [4]Division for the Establishment of Frontier Sciences of Organization for Advanced Studies, Tohoku University, Sendai, Japan. [5]Graduate School of Frontier Sciences, University of Tokyo, Kashiwa, Japan. [6]Department of Earth and Planetary Science, Graduate School of Science, University of Tokyo, Tokyo, Japan. [7]Faculty of Science, Tokyo University of Science, Tokyo, Japan. [8]Planetary Plasma and Atmospheric Research Center, Graduate School of Science, Tohoku University, Sendai, Japan. ✉e-mail: masunaga.kei@jaxa.jp

global/regional dust storms[14–16]. This is of interest because oxygen atoms, which are mainly produced by $CO_2$ photodissociation in the upper atmosphere, may be affected by the change in the vertical water distribution during major dust storms. This implies that some complexities exist when dust storms expand from the lower to the upper atmosphere. On this basis, comprehensive observations of hydrogen and oxygen in Mars' upper atmosphere, and dust and water vapor in Mars' lower atmosphere are needed.

In this paper, we analyze multi-spacecraft and space telescope observation data obtained during one of the major regional dust storms in Mars Year 33 (June 2015–May 2017) and show that the global abundance of hydrogen atoms in the upper atmosphere gradually increases while that of oxygen atoms decreases, responding to rapid variations of lower atmospheric conditions such as air temperature, dust and water ice opacities, and water vapor mixing ratio during the dust storm. We also show a presence of alternately and periodically varying hydrogen and oxygen abundances in the upper atmosphere during the dust storm, which is likely imposed by atmospheric waves propagating from the lower atmosphere.

## Results and Discussion
### A regional dust storm event in 2016
In early September 2016, a regional dust storm rapidly expanded on Mars, transporting a large amount of water vapor to altitudes greater than 60 km[12,17,18]. During this period, we observed the upper atmosphere using the Hisaki space telescope, which carried the Extreme Ultraviolet Spectroscope for Exospheric Dynamics instrument[19,20]. We analyzed the HI Ly-β, OI 1304 Å, and OI 1356 Å emissions to examine the variability of the total amount of hydrogen and oxygen atoms in Mars' upper atmosphere. We also obtained various measurements between the Mars surface and upper atmosphere using several instruments and spacecraft, including the Mars Climate Sounder[21] (MCS) on the Mars Reconnaissance Orbiter, the Spectroscopy for the Investigation of the Characteristics of the Atmosphere of Mars[22] (SPICAM) on the Mars Express (MEX), the Extreme Ultraviolet Monitor[23] (EUVM) and the Neutral Gas and Ion Mass Spectrometer[24] (NGIMS) on the Mars Atmosphere and Volatile Evolution (MAVEN), and the Rover Environmental Monitoring Station[25] (REMS) on the Mars Curiosity rover. These data analyses are described in detail in the Methods section.

On September 3, the regional dust storm started to expand ($L_S$~216°), during which dust opacity and air temperature increased rapidly (~3–5 days) in the 20–60 km altitude range (Figs. 1c, b, g, and f). Simultaneously, air pressure started to increase at all altitudes, although less rapidly than dust opacity and air temperature (Fig. 1n–p). Meanwhile, the $H_2O$ ice opacity began to decrease at the 20–40 km altitude after storm onset, whereas it swiftly increased at altitudes of 40–60 km (Fig. 1k, j). These observations indicate that because of the atmospheric warming caused by the lifted dust, the entire atmosphere expanded and the $H_2O$ ice cloud layer, usually present at altitudes of 10–20 km, was also raised to a higher altitude. Near the surface (0–20 km altitude), only small variations were observed for dust opacity, indicating an already dusty lower atmosphere. This was consistent with the fact that Mars was approaching the southern summer, during which the atmosphere is seasonally dusty, as shown in Supplementary Fig. 1.

### Comprehensive observations of the middle and upper atmosphere
Figure 2 shows the time series of middle and upper atmospheric observational data obtained via multiple spacecraft during the dust storm period. Figure 2j–m show the time series of the MCS air pressure, $H_2O$ ice opacity, air temperature, and dust opacity measurements averaged at an altitude range of 60–80 km, which are also shown in Fig. 1m, i, e, and a, respectively. In this range, according to the air pressure data, the atmosphere started to expand on the day of storm

onset and increased as rapidly as at the lower altitudes (Fig. 2j). The $H_2O$ ice opacity also rapidly surged for ~5 days after the storm onset (Fig. 2k), suggesting that the $H_2O$ ice cloud layer was lifted higher than 60 km. Meanwhile, the storm onset in dust opacity lagged a day in this altitude range, and its increasing speed, according to the peak opacity data (September 9 at 40–60 km and September 18 at 60–80 km), was also substantially delayed (~9 days). Figure 2n reveals the volume mixing ratio of water vapor at the four different altitude ranges observed by MEX/SPICAM. These measurements were assumed to be representative of the entire atmosphere, as they exhibited similar increasing tendencies between the northern and southern hemispheres at 20–60 km (Supplementary Fig. 2). Overall, the volume mixing ratio gradually increased with a slow increase in dust opacity. Notably, this ratio exceeded 100 ppm at an altitude of 60–90 km, which is a preferable condition for enhancing hydrogen escape from the upper atmosphere[9,26]. The rapid increase of $H_2O$ ice opacity and the slow increase of dust opacity in this altitude range may indicate that the lifted dust acted as cloud condensation nuclei. The high-altitude water vapor condensed on dust grains, and therefore, as the $H_2O$ ice opacity increased the dust opacity decreased during the formation of $H_2O$ ice clouds. The time lag in dust opacity may indicate that $H_2O$ ice clouds were being formed in this altitude range (60–80 km).

Figure 2a shows the daily solar ultraviolet irradiances of 305 Å, 1025 Å, and 1305 Å of the MAVEN EUVM l3 data. These data have ~5% variation during the period as a result of solar rotation and active regions on the solar surface. Figure 2g exhibits the time series of hydrogen (HI Ly-β) airglow in the upper atmosphere, as observed by the Hisaki telescope. Overall, Ly-β brightness started to increase gradually ~3 days after the storm onset and varied periodically over the period, revealing a behavior similar to the slow increase in dust opacity and water vapor mixing ratio at the 60–80 km altitude. Note that this variation pattern was different from that of the solar 1025 Å irradiance (Fig. 2a), suggesting a surge in hydrogen abundance in the upper atmosphere owing to high-altitude water vapor rather than an increased emission rate "g-factor." The disk brightness of the airglow approximately doubled over ~20 days, indicating that the column density of hydrogen atoms at least doubled, considering that Ly-β is an optically thick emission. The optical depth of Ly-β reaches near unity at 200 km altitude based on the H profile using the Chamberlain model[27] and exospheric parameters in ref. 28. Thus, our observation most likely reflects the hydrogen column density above 200 km altitude. The time scales of the hydrogen upper atmospheric response (~3 days) and its increase (~20 days) are comparable to those of vertical diffusion of hydrogen atoms dissociated from the high-altitude water at 60 km to the exosphere as shown by a photochemical model[26]. The time scale for the hydrogen increase also corresponded to a seasonal scale of $\Delta L_S$~15°. This value was close to the time lag between the Mars perihelion ($L_S = 251°$) at which airborne dust absorbs the solar flux the most (and thus dust storms would be the most common) and the nominal seasonal peak of hydrogen escape rates ($L_S$~270°)[7,13,29]. If the slow vertical diffusion of hydrogen atoms from an altitude of 60 km is common for every dust storm, the slow increase in hydrogen abundance in the upper atmosphere may explain the time lag between the Mars perihelion and the seasonal hydrogen escape peak every Mars year. Another explanation for the source of hydrogen atoms is that water vapor was transported to the altitude of the ionosphere. Although no mixing ratio was retrieved above 90 km with the SPICAM observations during the dust storm period, it is possible that water vapor was transported to the ionosphere at 100–150 km as recently observed by MAVEN[10] and the Trace Gas Orbiter[4] and that the water-origin hydrogen atoms increased in the upper atmosphere through collisions with ionospheric plasma[10]. Because most of the hydrogen atoms produced at this altitude diffuse upward to escape Mars and its time scale is short, hydrogen airglow would vary with the water mixing

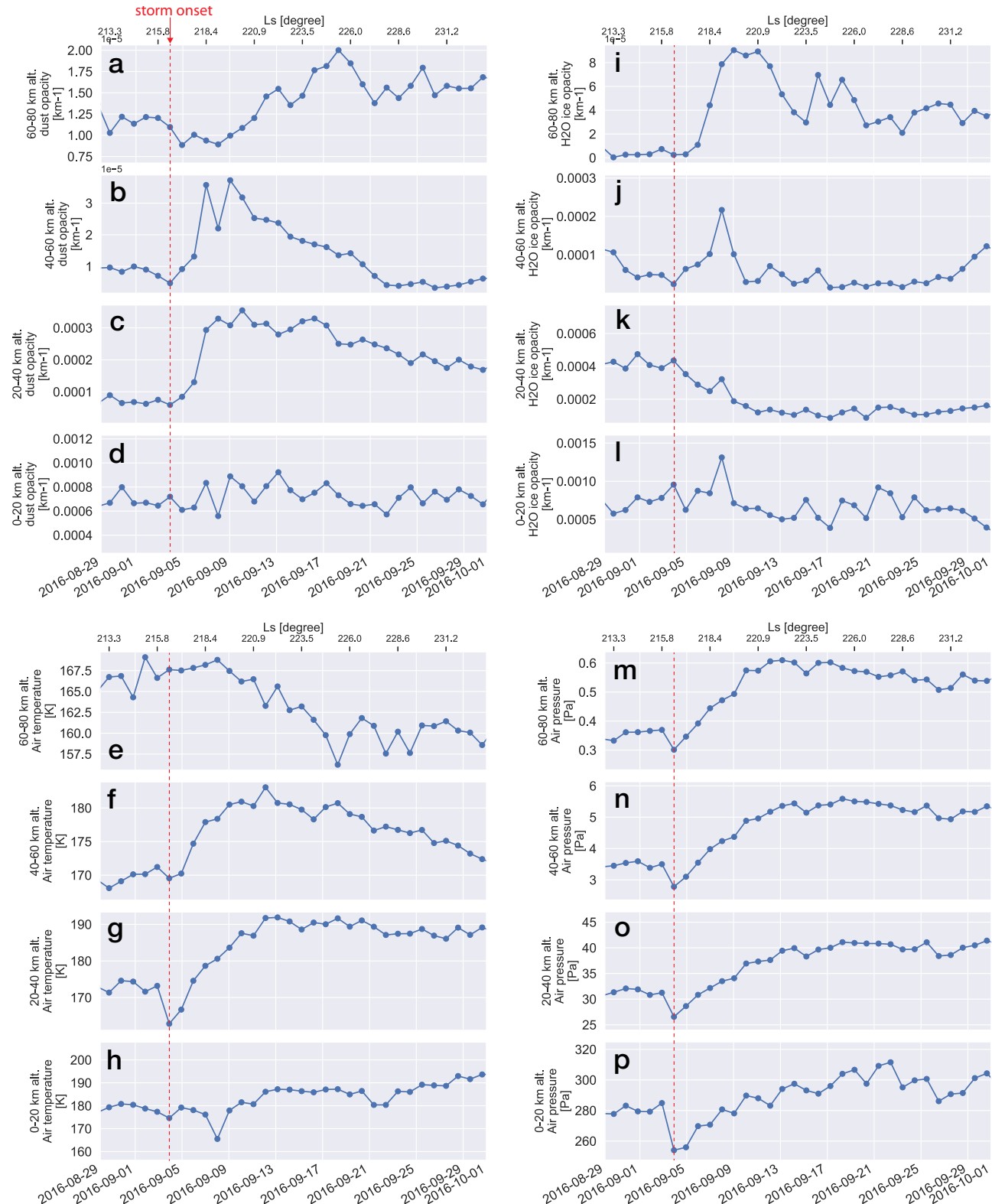

**Fig. 1 | Evolution of the regional dust storm in Mars Year 33.** Time series of (**a**–**d**) dust opacity, (**e**–**h**) air temperature, (**i**–**l**) $H_2O$ ice opacity, and (**m**–**p**) air pressure, as observed by the Mars Climate Sounder (MCS) limb observations at altitudes of 0–20 km, 20–40 km, 40–60 km, and 60–80 km. Each data point is the average value sampled at all latitudes and longitudes every Martian day (local time is limited between 11 h and 13 h). The red dashed lines indicate the dust storm onset date.

ratio. Our observations show that the increasing trends of the hydrogen airglow brightness above 200 km and water vapor mixing ratio up to 90 km are similar to each other (Fig. 2g, n). If the water vapor was transported to the altitude of the ionosphere and exhibited a similar variation to that of the lower altitude, hydrogen atoms may originate from the 100–150 km altitude.

The variation trend of the OI 1304 and OI 1356 brightness was similar to each other but different from that of the HI Ly-β airglow

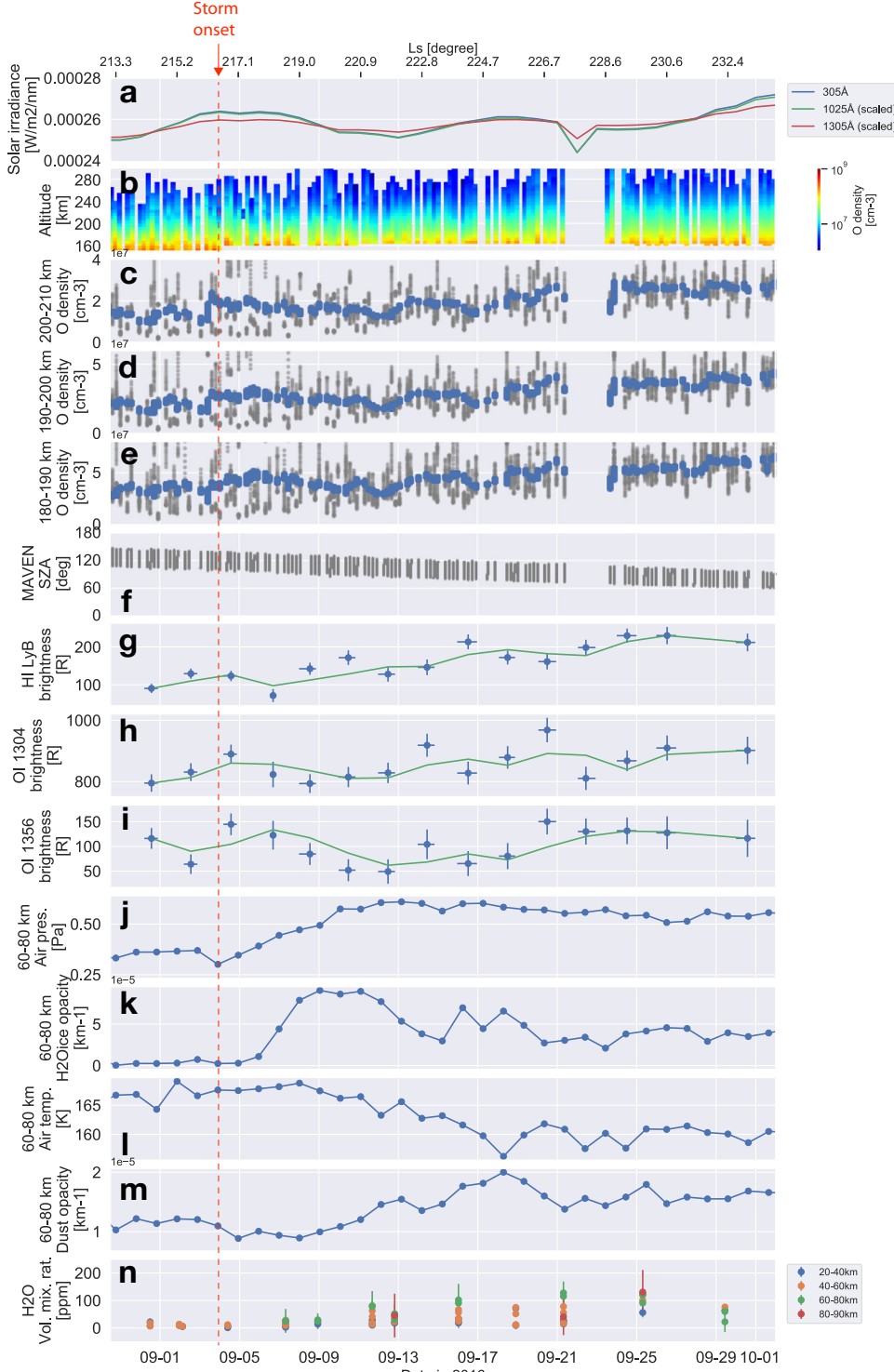

**Fig. 2 | Evolution of upper atmosphere during regional dust storm in Mars Year 33. a** Time series of the solar ultraviolet fluxes of Mars Atmosphere and Volatile Evolution Extreme Ultraviolet Monitor l3 data, **b** mean O number density profile at 150−300 km, and (**c**−**e**) O number density at 200−210 km, 190−200 km, and 180−190 km (grey dots for original data and blue dots for 1-day running mean), **f** solar zenith angle of MAVEN position, **g**−**i** disk brightness of hydrogen and oxygen airglow, as observed by the Hisaki telescope (the error bars denote 1σ standard deviation), **j** air pressure, **k** H₂O ice opacity, **l** air temperature, and **m** dust opacity at 60−80 km altitude, as observed by Mars Reconnaissance Orbiter/Mars Climate Sounder, and **n** volume mixing ratio of water vapor at four different altitudes observed by Mars Express/Spectroscopy for the Investigation of the Characteristics of the Atmosphere of Mars. The green lines in panels **g**−**i** indicate their rolling average data. The red dashed line indicates the onset date of the dust storm. To display the three solar irradiance data in the same panel (**a**), the solar 1025 Å and 1305 Å irradiance were scaled to the 305 Å irradiance value on August 31, 2016.

brightness (Fig 2h, i). The two oxygen emissions exhibited similar variations, likely because they mainly reflect variations of O abundance variations under the relatively small variations (-5%) of the solar 1304 Å and 305 Å fluxes (Fig. 2a) that affect the resonant scattering and photoelectron impact excitations of O atoms, respectively. OI 1304 is mainly excited by resonant scattering and is thus a direct indicator of the O column density, but it is an optically thick emission at Mars and its brightness is not linearly related to O column density[13,30]. On the other hand, because OI 1356 is an optically thin emission[30], its variations mainly reflect the total abundance of O atoms in the upper atmosphere. According to the rolling average data (green line in Fig. 2), the OI 1356 brightness rapidly declined by a factor of -3 in -6 days after the storm onset and then recovered to the original level after another -10 days. This suggested that the oxygen abundance in the upper atmosphere temporarily decreased by a factor of three after the dust storm. OI 1304 brightness exhibited a similar variation, but its time scale, including both decline and recovery, was shorter than that of OI 1356. This may result from OI 1356 only being excited by photoelectron impact with the peak intensity at -130 km, while OI 1304 was excited by the photoelectron impact at -130 km as well as by the resonant scattering of O atoms in a large altitude range by the solar 1304 flux[30]. Figure 2b shows NGIMS observations of O number density variation between 150–300 km. Variations of the O number density at 180–190 km, 190–200 km, and 200–210 km altitudes are also shown in Fig. 2c–e. Although the in-situ NGIMS measurements were not directly comparable with the entire disk observations of the Hisaki telescope, the O density variations in the three altitudes overall showed a similar trend to the O airglow variations. Although periodic variations were present, the O number densities decreased by a factor of -1.5 between the storm onset and September 12, and it subsequently increased to near the original level for -10 days. The similarity between the in-situ and entire disk measurements indicated that the variation of O abundance was a global feature in the Martian upper atmosphere. O depletion in the upper atmosphere during a global dust storm in 2018 has previously been reported[14,15], and our observation shows that even a regional dust storm globally affects the upper atmosphere of Mars.

One possible explanation for the O depletion is that rapid atmospheric heating below 60 km caused a rapid atmospheric expansion, forming $H_2O$ ice clouds even in the thermosphere that directly absorb O atoms. On Earth, direct uptake of O atoms has been observed near noctilucent clouds in the mesosphere[31,32]. Indeed, our observations show that $H_2O$ ice clouds rapidly formed at 60–80 km (Fig. 2k), and they may exist even above 100 km[10]. If the $H_2O$ ice clouds formed near 130 km altitude during this dust storm period, the direct uptake of O atoms could occur. After the O absorption, the $H_2O$ ice clouds may fall and eventually sublimate in the lower atmosphere, which indicates the transportation of O atoms to the lower atmosphere. Another explanation for the O depletion is that the loss of O was accelerated through odd hydrogen catalysis in the thermosphere[31–34]. The source of the odd hydrogen may be photolysis of water vapor or $H_2O$ ice clouds in the thermosphere[4,11,31,32]. At higher altitudes, the abundance of odd hydrogen decreases, and collisions for the catalysis would become ineffective. This is consistent with a greater decrease of O abundance observed at -130 km by OI 1356 airglow, than at 180–210 km by NGIMS. Another possibility is that dust storms may play a role in transporting not only water vapor but also dust-driven turbulence. High-altitude turbulence causes eddy diffusion that transports O atoms downward from the upper atmosphere[34,35]. In fact, the air temperature at 60–80 km altitude began to decrease gradually -4 days after the storm onset (Fig. 2l), which may demonstrate that $CO_2$ molecules were transported upward by atmospheric expansion while O atoms were transported downward, resulting in a larger $O/CO_2$ at an altitude of 60–80 km and radiative cooling of the $CO_2$ atmosphere[36,37].

## Alternating H and O abundances in the upper atmosphere

In addition to the gradual hydrogen rise and rapid oxygen decline in the upper atmosphere, the hydrogen and oxygen airglow exhibited periodic variations. Figure 3a, b show the residual brightness and variation percentages of the Ly-β and 1356 airglow with respect to their four-day rolling averages (green lines in Fig. 2g, i, respectively) as a function of time. Clear periodic variations appear both in the hydrogen and oxygen residual brightness, exhibiting an alternating relationship. Using the Lomb-Scargle periodogram method[38,39], we obtained a periodicity of 6.6 days in Ly-β (Fig. 3d). A peak was observed at -8 days for the OI 1356 emission, although this peak was smaller than the confidence level. Yet, a 6.8-day periodicity was observed in NGIMS O density data at 180–210 km altitude (Supplementary Fig. 3). The amplitude of the airglow variations was 20-50% (Fig. 3b), suggesting that the global abundances (i.e., column density) of H and O atoms alternately varied by 20–50% every 7–8 days. Figure 3c shows the Mars Curiosity rover observations of air pressure and air temperature averaged on the dayside. Measurements demonstrate that air pressure gradually increased over the period, which was consistent with that observed by MCS. Conversely, air temperature exhibited distinct periodic variations ranging between 250 K and 270 K, demonstrating a periodicity of -6 days, as determined via the periodogram method (Fig. 3e). These features were consistent with those of baroclinic waves usually observed in this season[40,41]. This periodicity was reasonably comparable to the airglow periodicity, given that the rover and space telescope measurements were based on a single point on Mars and the global view of the entire disk, respectively. This implies that the baroclinic waves propagated horizontally and vertically, modulating the atmospheric circulation to affect dust and water transportations from the lower to the upper atmosphere, which periodically affected the formation of $H_2O$ ice, odd hydrogen productions, or downward transportations and alternately changed H and O abundances in the upper atmosphere.

Our findings offer direct evidence for a coupling between the Martian lower and upper atmospheres on the dayside via a major regional dust storm and coincident atmospheric waves. The dust storm event investigated in this study played a role in increasing the hydrogen abundance in the upper atmosphere by a factor of -2 over 20 days, but temporarily reduced oxygen abundance by a factor of 3 over 6 days. In addition, vertically propagating baroclinic waves were found to modulate dust, water, and atmospheric circulation, resulting in alternate oscillations of hydrogen and oxygen abundances in the upper atmosphere. Water ice clouds, odd hydrogen catalysis, or dust-driven turbulence as well as high-altitude water may be fundamental in causing these variations in hydrogen and oxygen atmospheres. If these are common processes for every dust storm, more hydrogen atoms may equate to fewer oxygen atoms in the upper atmosphere. This potentially stipulates that with increasing hydrogen escape, oxygen will less efficiently escape from Mars. Thus, dust storms play a role in oxidizing the atmosphere, at least on present-day Mars where dust storms seasonally occur. If dust storms have repeatedly occurred and controlled the hydrogen and oxygen escape, past Mars may have had a less oxidizing atmosphere than today, although there are many uncertainties in the history of Mars atmospheric and dust storm conditions. Hence, our findings provide insights regarding the Martian history of water loss and its redox state, which is crucial for a habitable environment.

## Methods
### Hisaki Space Telescope dataset
In this study, we analyzed the extreme ultraviolet (EUV) spectra obtained from the Hisaki Space Telescope[19,20]. The Martian upper atmosphere was observed from the end of August to the end of September 2016, during which a regional dust storm occurred. We obtained Mars EUV spectra every two days (except for September 28).

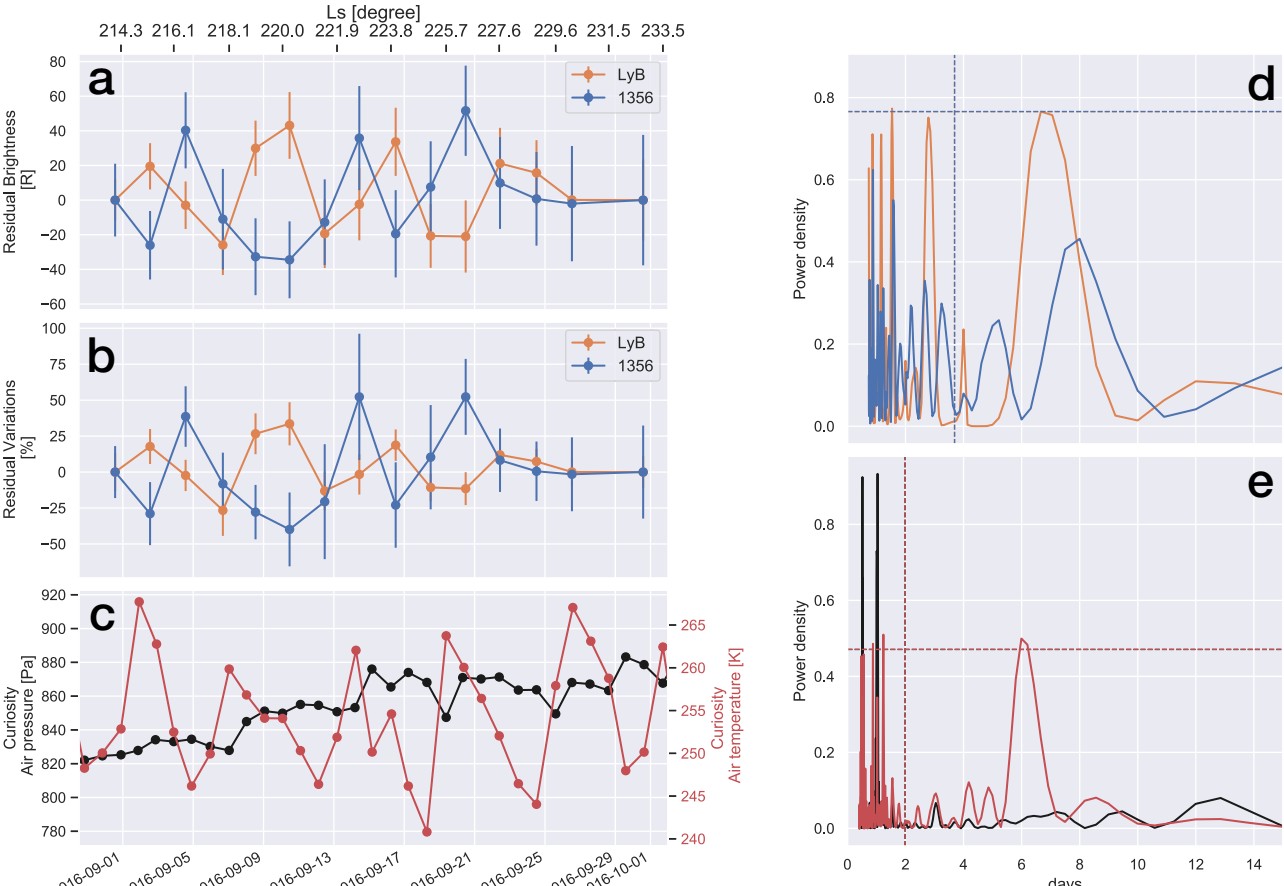

**Fig. 3 | Variations in hydrogen and oxygen airglow, air temperature, and air pressure during dust storm.** Time series of **a** Ly-β and **b** OI 1356 airglow residual brightness according to rolling average data (the error bars denote 1σ standard deviation); and **c** average air pressure and temperature at local times of 6 h and 18 h, as observed by Rover Environmental Monitoring Station (REMS). Normalized power spectrum of **d** airglow emissions and **e** air temperature and pressure calculated using the Lomb-Scargle periodogram method. Vertical and horizontal dashed lines correspond to Nyquist frequency periods and confidence levels of 90% and 99%, respectively.

Examples of daily-averaged spectra of HI Ly-β (1025 Å), OI 1304 Å, and OI 1356 Å airglow are shown in Supplementary Figs. 4–6. We calculated the average disk brightness and its error using the following equations:

$$B_{Mars} = (C_{tot} - C_{BG}) \cdot G \cdot \frac{\Omega}{\Omega_d} \quad (1)$$

$$\triangle B_{Mars} = \sqrt{\triangle C_{tot}^2 + \triangle C_{BG}^2} \cdot G \cdot \frac{\Omega}{\Omega_d} \quad (2)$$

where $C_{tot}$ and $C_{BG}$ are count rates of the entire Mars disk and background observations (integrations of data in panels d and e in Supplementary Figs. 4–6, respectively) and $\triangle C_{tot}$ and $\triangle C_{BG}$ are their Poisson errors (standard deviations). G is the geometric factor converting count rates to Rayleigh. $\Omega$ and $\Omega_d$ are solid angles of a single pixel and the illuminating are of the Mars disk, respectively, and their ratio is calculated as

$$\frac{\Omega}{\Omega_d} = \frac{\triangle\varphi\triangle\theta}{\pi(d/2)^2 N_{R_M}^2 p} \quad (3)$$

where $\triangle\varphi$ and $\triangle\theta$ are the angular plate scales in the spectral and slit directions, corresponding to 10 arcsec and 4.2 arcsec, respectively. The value d is the apparent diameter of Mars and p is the fraction of the illuminating area. $N_{R_M}$ is selected to be unity for OI 1304 and OI 1356 and two for HI Ly- β to scale the size of the illuminating area. Following the above data processing method, we calculated the 2-$R_M$ ($R_M$

denotes a Mars radius) disk brightness of the hydrogen Ly-β airglow and the 1-$R_M$ disk brightness of the oxygen 1304 Å and 1356 Å airglow every other day. We discarded data affected by cosmic rays or EUV stars, and only used those obtained when the telescope orbited inside the shadow of Earth (defined as the local time between 20 h and 4 h) to reduce the geocoronal effect. We then employed a data selection method to increase data quality by removing pointing errors[13]. Finally, using these data, we studied HI Ly-β (1025 Å), OI 1304 Å, and OI 1356 Å airglow variations every two days during the dust storm event. Note that HI Ly-α data was not used in this study because the detector's sensitivity near 1216 Å significantly degraded in the middle of the mission due to its constant exposure to the extremely strong Ly-α emissions of geocorona and planetary upper atmospheres such as Venus[42–44] and Jupiter[45]. So, Ly-α data is currently available only at the beginning of the mission[46]. In-flight calibration for this issue is currently ongoing for the future use of the whole Ly-α dataset.

### Mars Reconnaissance Orbiter (MRO)/Mars Climate Sounder (MCS) dataset

MRO is orbiting Mars since March 2006 to study the geology and climate of Mars[47], and the MCS instrument has measured thermal emission of the Martian atmosphere[21]. In this study, we used a dataset of dust opacity, $H_2O$ ice opacity, air temperature, and air pressure measurements retrieved by MCS, which were stored in the MRO/MCS derived data records Version 5 in the NASA Planetary Data System. To study the dynamics of Mars' dayside lower and middle atmospheres, these measurements were averaged within the local time of 11 h and

13 h for every 20 km altitude between 0 km and 80 km every Martian day. The dataset at an altitude of 0–20 km during Mars Year 33 is shown in Supplementary Fig. 1, and MCS geometry information during Hisaki's observation period is shown in Supplementary Fig. 7.

## Mars Atmosphere and Volatile Evolution (MAVEN)/Solar Extreme Ultravioet Monitor (EUVM) and Neutral Gas and Ion Mass Spectrometer (NGIMS) dataset

MAVEN is orbiting Mars since September 2014 to investigate the Martian atmospheric escape processes controlled by solar and atmospheric forcings[48]. In this study, we used the solar ultraviolet irradiance of EUVM level 3 data[49], obtained from the Laboratory for Atmospheric and Space Physics Interactive Solar Irradiance Data Center. We selected the 305 Å line as a proxy for photoionization of the $CO_2$ atmosphere to produce photoelectrons in the Martian ionosphere[50] and the 1025 Å and 1305 Å lines as sources of resonantly scattered HI Ly-β and OI 1304 Å airglow[13], respectively. To display the three irradiance data in the same panel in Fig. 2a, the solar 1025 Å and 1305 Å irradiances were scaled to the 305 Å irradiance value on August 31, 2016.

We also used O number density of NGIMS level 2 data. In this study, we only used inbound measurements near periapsis (150–300 km altitude range). Supplementary Fig. 3a shows binned (5 km by 5 h) O densities between 150–300 km, which is identical to Fig. 2b. Supplementary Fig. 3b–d show O density variations of three different altitude ranges (180–190 km, 190–200 km, and 200–210 km), which are also identical to Fig. 2c–e. To analyze their average diurnal variations, we calculated their running mean values using a 1-day window (blue dots). Supplementary Fig. 3e–h show MAVEN geometry data. We can see that MAVEN positions developed from the nightside to the dayside across the terminator in this period.

## Mars Express (MEX)/Spectroscopy for the Investigation of the Characteristics of the Atmosphere of Mars (SPICAM) dataset

MEX is orbiting Mars since December 2003 and has collected comprehensive dataset, including atmospheric water contents with the SPICAM instrument[22]. In this study, we used the volume mixing ratio of water vapor in the Martian atmosphere retrieved from the SPICAM measurements by ref. 18. Depending on their tangential attitude, we divided the measurements into four altitude ranges (20–40 km, 40–60 km, 60–80 km, and 80–90 km), as shown in Fig. 2n. Supplementary Fig. 2 also shows the observation coverage of the SPICAM. In 20–60 km altitude ranges, SPICAM measurements cover both northern and southern hemispheres, whereas they are only derived from the southern hemisphere in the 60–90 km range.

## Curiosity/ Rover Environmental Monitoring Station (REMS) dataset

The Curiosity rover landed in the Gale crater at the longitude of 137.4°E and latitude of 4.6°S in early August 2012[51]. Since then, the rover has performed various experiments and observations to study habitable environment on Mars, including its meteorology with the REMS instrument[25]. When the Hisaki telescope observed the Martian upper atmosphere, the rover was traveling over a region called Murrays Buttes, ~8 km southwest of the landing site, and the REMS instrument monitored atmospheric conditions. In this study, we used a dataset of air temperature and pressure of the REMS-reduced data stored in the NASA Planetary Data System. We averaged these measurements within the local time of 6 h and 18 h to focus on their dayside variations as shown in Fig. 3c.

## Data availability

The Hisaki level 2 data are publicly available in the JAXA Data Archives and Transmission System (DARTS, https://data.darts.isas.jaxa.jp/pub/ hisaki/euv/l2/). The Mars Atmosphere and Volatile Evolution Extreme Ultraviolet Monitor l3 data are publicly available at the LASP Interactive Solar Irradiance Data Center (https://lasp.colorado.edu/lisird/). Neutral Gas and Ion Mass Spectrometer l2 data are publicly available at MAVEN Science Data Center (https://lasp.colorado.edu/maven/sdc/ public/pages/datasets/ngims.html). Mars Climate Sounder and Curiosity data are publicly available in the NASA Planetary Data System (https://atmos.nmsu.edu/data_and_services/atmospheres_data/MARS/ mcs.html and https://atmos.nmsu.edu/PDS/data/mslrem_1001/, respectively). The Mars Express/Spectroscopy for the Investigation of the Characteristics of the Atmosphere of Mars data derived from ref. 16 are stored at https://data.mendeley.com/datasets/vx4gks6bx7/1, and we would like to thank the authors for making their dataset publicly available. The Hisaki disk average airglow brightness data generated in this study is provided in Supplementary Data 1.

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

## Acknowledgements

K.M. is supported by JSPS KAKENHI (Grant numbers JP21K20387 and JP22K03708) and the JAXA Hisaki and MMX projects. N.T. is supported by JSPS KAKENHI (Grant numbers JP18H05439, JP18KK0093, JP19H00707, JP20H00192, and JP22H00164). N.Y. is supported by JSPS KAKENHI Grant number JP21J13710 and the International Joint Graduate Program in Earth and Environmental Sciences, Tohoku University (GP-EES). Y.N. is supported by JSPS KAKENHI Grant Number JP22J14954 and the International Joint Graduate Program in Earth and Environmental Sciences, Tohoku University (GP-EES).

## Author contributions

K.M. conceived the study and analyzed the datasets of the Hisaki telescope and multiple spacecraft, and wrote the paper. N.T., N.Y., Y.N., T.K., K.Y., Y.S., H.N., and T.U. contributed to the scientific interpretation of the results. K.Y., G.M., F.T., and T.K. contributed to the calibration of the Hisaki data. A.Y. led the operation of the Hisaki space telescope for Mars observation. I.Y. directed the Hisaki mission as the principal investigator. All authors participated in the discussion of the results of this paper.

## Competing interests

The authors declare no competing interests.

## Additional information

**Correspondence and requests** for materials should be addressed to Kei Masunaga.

