## [Peer Review File · Nature Communications]

REVIEWER COMMENTS

Reviewer #1 (Remarks to the Author):

Alternate oscillations of Martian hydrogen and oxygen upper atmospheres during a major dust storm

Masunaga et al. submitted to Nature Communications

This paper reports observations of the upper atmosphere of Mars with the Hisaki mission that shows the change in densities of H and O during a PEDE dust storm. Data from other spacecraft are shown to track the changes in the atmosphere and progression of the dust storm. The authors conclude that there are contrasting changes, with the H density increasing and O density decreasing, and that there are oscillations in these densities consistent with the time scale for baroclinic waves in the atmosphere. They go on to discuss the change in oxidation state of the overall atmosphere due to an imbalance of H and O loss to space.

Overall the paper is well written and interesting to read. The presentation of Hisaki data is new to the study of the 2018 PEDE, whereas other data on this event have been previously published. The identification of the oscillations appears to be solid, and the discussion of the implications is interesting. The identification of an increase in H density supports earlier reports of this, and the discussion of possible changes in oxidation state of the atmosphere is interesting if not overly well established. One feature of the Hisaki data is that the measurements give a global average of the emissions, whereas other spacecraft at Mars get point by point data. This can be an advantage in analyzing the PEDE event, which was a global dust storm, in deriving the global atmospheric response. One question I have is that in the Elrod paper there appear oscillations in the NGIMS O density values between LS 190 - 240, and I wonder if the authors could analyze those data to further develop their presentation of this phenomenon. If the oscillations are real they would be expected to appear in both datasets, and the NGIMS data have much better time sampling.

One problem with this paper is that the changes in O density are not convincing. In Fig. 2 the slow increase in H is clearly seen, whereas for the O 1304 and 1356 emissions there is not a clear long-term trend either up or down. The main feature of those emissions is the oscillation that the authors analyze, which is well established from the Hisaki data. The 1304 emission shows a long term slow increase, with a high value just after the PEDE commencement and then a few low points. The problem is that these changes are 2-3 sigma from the plotted error bars, consistent with the discussed oscillation (i.e. not unique to the PEDE), and not supported by a long term decrease. The decrease in O during the PEDE is well established from the data published from the MAVEN NGIMS experiment, and the authors could refer to that for support for changes in O density and to better characterize those changes.

For changes in the O density, the 1304 emission is a better indicator of the O density than 1356. The 1304 emission is produced mainly by resonant scattering of solar emission while the 1356 is produced mainly by electron collisions, and while the solar 1304 flux can be measured the electron density is not well known. In this case the 1304 can be related to O density while more information is needed to derive density from 1356.

The Nature editor has asked for an evaluation of the significance of this work, and the extent to which it is new. The main results from measurements of the 2018 PEDE have been published before, and several papers are referenced in this work. The new results presented in this paper are mainly the Hisaki data and also more data from MCS (I am not aware of how much of this data has been previously published). Overall I have to describe this paper as an incremental improvement on our understanding of the atmospheric response to the 2018 PEDE, rather than a ground-breaking new direction of research.

Reviewer #2 (Remarks to the Author):

The manuscript titled "Alternate oscillations of Martian hydrogen and oxygen upper atmosphere during a major dust storm" by Masunaga et al. presented multi-instruments data to show the effect of a dust storm on Martian atmosphere evolution.

The results are compelling and significant in the field of Martian atmosphere evolution given the evidence of dust-driven variability in the upper atmosphere and coupling between the lower atmosphere via the wave activities. The use of wave activity and observed oscillations in the emissions to deduce the time scale of dust impact on the atmospheric constituents is very novel and noteworthy.

I would recommend the publication of this study after some clarification.

1. The authors have referenced their previous study for Hisaki observations and data reduction. However, their published study only focused on OI 1304 and H Lyman Beta emission but did not mention OI 1356 A emission. Given the low disk brightness of OI 1356, it would be imperative to show the data for this emission in this study.

2. Line 125-126: Authors have indicated linear relation between OI 1356 brightness and Oxygen density, however electron impact on CO₂ also contributes to this emission. Please explain.

Reviewer #3 (Remarks to the Author):

This study describes the analysis of multi-spacecraft observations of the atmosphere of Mars in order to investigate the response of oxygen and hydrogen in the upper atmosphere as a consequence of dust storms in the lower and middle atmospheres. This study is in agreement with other studies showing that dust storms cause an increase in the exospheric H density and a decrease in the O density. In addition, the observations analysed in this study show an oscillating and anti-correlated pattern in the H and O brightness, which are attributed atmospheric waves being propagated from the surface. The results of this study have implications to understand the history of the Martian atmosphere and the role of atmospheric escape, and may be published after addressing the following comments.

Abstract

Line 25: slow dust transportation compared to atmospheric expansion  The meaning of this sentence is unclear

Line 28: oxygen decrease via downward transportation  the main text appears to indicate that there are different scenarios that could explain the decrease in the O abundance in the exosphere.

Main text

Lines 42-45: The ascent of water to high altitudes is not only driven by dust storms, but also occurs every year close to during southern spring and summer (close to perihelion), even in the absence of major dust storms. Water increase during perihelion/dust storms is shown by Belyaev et al. (2021), and Alday et al. (2021) showed the expected seasonal variations in the H₂O photolysis (i.e., H production), which are mainly driven by the Mars-Sun distance even in the absence of dust storms.

Lines 48-50: While H₂O is the main reservoir of H in the atmosphere, the O in H₂O just represents a very small part of the O reservoir in the atmosphere (95% CO₂). Are the variations in the O upper atmosphere during dust storms caused by the H₂O increase at high altitudes, or by the expansion of the whole atmosphere due to the intensified warming?

Line 68: rapidly  rephrase with a more specific term. For example, something like: on a matter of X days

Lines 70-74: From those plots it indeed looks that the cloud base was not situated at the 10-20 km level, which does not experience any change because of the dust storm. It looks as if the cloud layer was in the 20-40 km range, and the intensified warming because of the storm raised this level to the 60-80 km level. The increase in H₂O at 40-60 km is very sharp, but also brief and transient, the increase in the 60-80 km level appears steadier. I think the interpretation is that the intensified warming in the middle atmosphere prevents the clouds to form at these lower levels, and water ice clouds are formed instead at much higher altitudes. I would consider including the 60-80 km level in Figure 1 to help this discussion.

Line 84: demonstrate  show the time series...

Line 87: as rapidly, similar to its behavior at lower altitudes  As rapidly as at the lower altitudes?

Lines 89-94: What about dust acting as cloud condensation nuclei? The water will condense on the dust grains and therefore as the water ice opacity increases the dust opacity will decrease. Therefore, I am not sure that the interpretation of strong advection up to 60 km and then a decline above this altitude is entirely correct. The lag in the dust opacity might just indicate that water ice clouds are being formed at those altitudes.

Line 97: There is just one data point that exceeds 100 ppm at 40-60 km. In addition, the coverage of the SPICAM H₂O profiles is quite sparse. It should be addressed in the text how representative the SPICAM profiles are for representing the whole atmosphere.

Lines 98-100: This interpretation that the increase in H₂O in the middle atmosphere comes from a release of water by dust particles needs proof, and it might not be correct. The warmer temperatures due to the dust heating prevent the formation of water ice clouds, which essentially confine water vapour below the cloud level. In addition, the intensified atmospheric transport during dust storms increase advection and raise water to the upper layers (e.g., Heavens et al. 2019, GRL; Shaposhnikov et al. 2019, GRL).

Line 106: its  this

Lines 109-110: Is the Ly-beta emission optically thin? This will determine whether there is a linear relation between the density and the brightness.

Line 120: was much larger  provide some estimates of the % of variation.

Line 122: Are the oxygen emissions optically thin? See comment above.

Line 125: in accordance with the observed atmospheric expansion  is the increase or the decrease of O in accordance with the atmospheric expansion?

Lines 125-126: The decrease in the O densities during dust storms has been reported before and should be acknowledged in the text.

Lines 128: Does turbulence only affect the O transport, or all gaseous species?

Line 137: 130 km is the peak intensity  this should be mentioned before making the calculation of the downward oxygen transportation.

Lines 126-146: It seems that several scenarios are suggested that could explain the decrease of O in the upper atmosphere. I suggest to re-organise this long paragraph to clearly highlight how these different scenarios would work.

Lines 148-149: The solar flux in Figure 2a also appears to oscillate. Is the frequency of the solar flux variations similar to the variations in the H or O brightness? That would not explain the anti-correlation between the O and H oscillations, though.

Line 150: Are these variations also observed in the 1304 O brightness?

Lines 151-152: Here the analysis focuses on the dust storm. Are these oscillations also present in non-dusty periods?

Lines 164-169: As mentioned in the text, the surface measurements are representative of a very specific location of the surface, while the airglow measurements encircle the whole planet, making their relation not that straightforward. Are similar oscillating patterns like these observed in the MCS temperature data at different altitude levels? The MCS dataset is likely more representative of the global behaviour of the atmosphere and will indicate how these waves propagate.

Line 171: Is the downward O transportation the reason of the decrease in the O brightness? From the paragraph before it seemed there could be several explanations.

Line 178-189: Reference 4 (Chaffin et al. 2021) is for a regional dust storm in MY34. Stone et al. (2021) (reference 5) also indicate variations in the water abundance at high altitudes in a time scale of about 2 days for the global dust storm in MY34.

Lines 181-184: This should be rephrased. While the transport timescale might be like that, H atoms formed at 20 km will mostly not make it to 130 km, since the timescale for photochemical H loss in the lower atmosphere is much shorter (Gonzalez-Galindo et al. 2005, JGR).

Figure 1

In my opinion, it would be more intuitive to put the panels in the reversed order: 0-20 km and the highest altitude range the uppermost panel, as if they were all sharing the same y-axis.

Do the plotted lines include errorbars? They are averages, but they could include the standard deviation of all the averaged points to give an idea of the planet-wide variability.

I think this figure should also include the 60-80 km range for the plotted parameters to easily follow the discussion and interpretation of the climatological parameters.

Figure 2

The flux at 305A is not mentioned at all in the text.

As mentioned, some of the panels for the 60-80 km range could be included instead in Figure 1.

It must be indicated in the caption what the arrows represent.

Figure 3

Include errorbars in the Residual brightness.

Include the 1304 A emission and search for similar patterns in MCS data.

Methods

Lines 333-340: I guess that this dataset relies on the limb-viewing observations. Indicate the number of observations and coverage used in this study.

Lines 350-351: This should be included in the caption of Figure 2.

Line 355: has collected a comprehensive...

Line 353: For SPICAM please also indicate the coverage and number of observations.

RESPONSE TO REVIEWERS' COMMENTS

Reviewer #1 (Remarks to the Author):

Alternate oscillations of Martian hydrogen and oxygen upper atmospheres during a major dust storm

Masunaga et al. submitted to Nature Communications

This paper reports observations of the upper atmosphere of Mars with the Hisaki mission that shows the change in densities of H and O during a PEDE dust storm. Data from other spacecraft are shown to track the changes in the atmosphere and progression of the dust storm. The authors conclude that there are contrasting changes, with the H density increasing and O density decreasing, and that there are oscillations in these densities consistent with the time scale for baroclinic waves in the atmosphere. They go on to discuss the change in oxidation state of the overall atmosphere due to an imbalance of H and O loss to space.

We would like to thank the reviewer for the helpful comments after the thorough reading of this manuscript. We have considered all the suggested comments and revised our manuscript.

Upon your request, we added MAVEN/NGIMS data to the analysis, which improved our interpretation and discussion. We also noticed that there were SPICAM water mixing ratio data (3 data points) at 80-90 km and added them to improve the discussion. Furthermore, we revised the texts and figures based on all reviewer comments. We would like to emphasize that, nevertheless, these changes did not affect our conclusion.

We describe our point-to-point responses to each comment below with the original comments in black and our response in blue. The line number in the reply refers to the revised manuscript with a track change.

Overall the paper is well written and interesting to read. The presentation of Hisaki data is new to the study of the 2018 PEDE, whereas other data on this event have been previously published. The identification of the oscillations appears to be solid, and the discussion of the implications is interesting. The identification of an increase in H density supports earlier reports of this, and the discussion of possible changes in oxidation state of the atmosphere is interesting if not overly well established. One feature of the Hisaki data is that the measurements give a global average of the emissions, whereas other spacecraft at Mars get point by point data. This can be an advantage in analyzing the PEDE event, which was a global dust storm, in deriving the global atmospheric response. One question I have is that in the Elrod paper there appear oscillations in the NGIMS O density values between LS 190 - 240, and I wonder if the authors could analyze those data to further develop their presentation of this phenomenon. If the oscillations are real they would be expected to appear in both datasets, and the NGIMS data have much better time sampling.

Please let us clarify that we are not studying the 2018 PEDE event in this paper but a regional dust storm event (A-storm) in 2016. Nevertheless, we analyzed NGIMS data obtained in our observation period and revised the paper (lines 177-188).

Based on our NGIMS data analysis (see Figs. 2b-e), O densities in three different altitudes overall showed similar variations to oxygen airglow variations. The O density exhibited a decrease by a factor of ~ 1.5 between the storm onset and Sep 12. After that, it gradually recovers to the original level. We also studied periodicities of the NGIMS data using the periodogram method (Supplementary Fig. 2), and we

found a 6.8-day periodicity. These consistencies between the in-situ NGIMS and entire disk Hisaki measurements suggest that the observed O variations were a global feature in the upper atmosphere of Mars.

One problem with this paper is that the changes in O density are not convincing. In Fig. 2 the slow increase in H is clearly seen, whereas for the O 1304 and 1356 emissions there is not a clear long-term trend either up or down. The main feature of those emissions is the oscillation that the authors analyze, which is well established from the Hisaki data. The 1304 emission shows a long term slow increase, with a high value just after the PEDE commencement and then a few low points. The problem is that these changes are 2-3 sigma from the plotted error bars, consistent with the discussed oscillation (i.e. not unique to the PEDE), and not supported by a long term decrease. The decrease in O during the PEDE is well established from the data published from the MAVEN NGIMS experiment, and the authors could refer to that for support for changes in O density and to better characterize those changes.

As described above, we found a factor of ~1.5 decreases in O density in NGIMS measurements at 180-210 km. However, since both Hisaki O airglow and NGIMS O density recover to the original level after the decrease, we concluded that O abundance “temporarily” decreased in the upper atmosphere. Several possibilities to cause the O depletion were raised as revised at lines 190-209. We also added a discussion regarding the O decrease in the PEDE event in 2018 (lines 186-188).

For changes in the O density, the 1304 emission is a better indicator of the O density than 1356. The 1304 emission is produced mainly by resonant scattering of solar emission while the 1356 is produced mainly by electron collisions, and while the solar 1304 flux can be measured the electron density is not well known. In this case the 1304 can be related to O density while more information is needed to derive density from 1356.

The 1304 emission is not only excited by resonant scattering but also by photoelectron impact in the ionosphere. Therefore, this emission reflects the column density of cold oxygen atoms excited by photoelectron impact near 130 km altitude (ionospheric peak altitude) as well as hot oxygen atoms excited by resonant scattering at higher altitudes. In addition, the 1304 emission is optically very thick, so its brightness is not linearly correlated to column density. On the other hand, the 1356 emission is mainly excited by photoelectron impact on O atoms and this emission is spin-forbidden and thus optically thin. Thus, the 1356 emission rather provides information on oxygen column density near 130 km altitude (ionospheric peak altitude).

The Nature editor has asked for an evaluation of the significance of this work, and the extent to which it is new. The main results from measurements of the 2018 PEDE have been published before, and several papers are referenced in this work. The new results presented in this paper are mainly the Hisaki data and also more data from MCS (I am not aware of how much of this data has been previously published). Overall I have to describe this paper as an incremental improvement on our understanding of the atmospheric response to the 2018 PEDE, rather than a ground-breaking new direction of research.

Again, please let us clarify that we are not studying the 2018 PEDE event in this paper but a regional dust storm event (A-storm) in 2016. Although a PEDE event produces a drastic change in the Martian atmospheric condition, it only occurs every 3–4 Mars Years (6-8 Earth years) on average. On the other hand, major regional dust storms (A-, B- and C- storms) occur 3 times every Mars Year. Even though a regional dust storm does not cover the entire surface of Mars, our observation shows that it affects the

upper atmosphere globally. If the dust- and wave-coupling effects on the upper atmosphere, and the atmospheric oxidation process that we suggested in this paper are common features in all regional dust storms, it is possible that the Martian atmosphere has been oxidized every Mars Year over a long period of Mars' history. In this context, regional dust storms provide a significant impact on the atmospheric evolution of Mars, and thus, our findings are crucial for understanding habitable environments on Mars.

Reviewer #2 (Remarks to the Author):

The manuscript titled "Alternate oscillations of Martian hydrogen and oxygen upper atmosphere during a major dust storm" by Masunaga et al. presented multi-instruments data to show the effect of a dust storm on Martian atmosphere evolution.

The results are compelling and significant in the field of Martian atmosphere evolution given the evidence of dust-driven variability in the upper atmosphere and coupling between the lower atmosphere via the wave activities. The use of wave activity and observed oscillations in the emissions to deduce the time scale of dust impact on the atmospheric constituents is very novel and noteworthy.

I would recommend the publication of this study after some clarification.

We would like to thank the reviewer for the helpful comments after the thorough reading of this manuscript. We have considered all the suggested comments and revised our manuscript.

Upon a request from another referee, we added MAVEN/NGIMS data to the analysis, which improved our interpretation and discussion. We also noticed that there were SPICAM water mixing ratio data (3 data points) at 80-90 km and added them to improve the discussion. Furthermore, we revised the texts and figures based on all reviewer comments. We would like to emphasize that, nevertheless, these changes did not affect our conclusion.

We describe our point-to-point responses to each comment below with the original comments in black and our response in blue. The line number in the reply refers to the revised manuscript with a track change.

1. The authors have referenced their previous study for Hisaki observations and data reduction. However, their published study only focused on OI 1304 and H Lyman Beta emission but did not mention OI 1356 A emission. Given the low disk brightness of OI 1356, it would be imperative to show the data for this emission in this study.

We agree with the reviewer. We've decided to add examples of the OI 1356 spectrum as well as OI 1304 and HI Ly-beta in the Supplementary Figure.

2. Line 125-126: Authors have indicated linear relation between OI 1356 brightness and Oxygen density, however electron impact on CO₂ also contributes to this emission. Please explain.

According to MAVEN/IUVS observations by Ritter et al. (2019), JGR, 10.1029/2019JA026669, the main contributor (95%) for the 1356 emission is electron impact on O, and electron impact on CO₂ is a neglected source (5%). Additionally, the 1356 emission is optically thin. Therefore, we interpret the 1356 brightness as being correlated to oxygen (column) density.

Again, we would like to thank the reviewer for reading our manuscript carefully and giving helpful comments to make this paper's quality better. We hope that the revised manuscript is suitable for publication in Nature Communications.

*Sincerely,
Kei Masunaga*

Reviewer #3 (Remarks to the Author):

This study describes the analysis of multi-spacecraft observations of the atmosphere of Mars in order to investigate the response of oxygen and hydrogen in the upper atmosphere as a consequence of dust storms in the lower and middle atmospheres. This study is in agreement with other studies showing that dust storms cause an increase in the exospheric H density and a decrease in the O density. In addition, the observations analysed in this study show an oscillating and anti-correlated pattern in the H and O brightness, which are attributed atmospheric waves being propagated from the surface. The results of this study have implications to understand the history of the Martian atmosphere and the role of atmospheric escape, and may be published after addressing the following comments.

We would like to thank the reviewer for the helpful comments after the thorough reading of this manuscript. We have considered all the suggested comments and revised our manuscript.

Upon a request from another referee, we added MAVEN/NGIMS data to the analysis, which improved our interpretation and discussion. We also noticed that there were SPICAM water mixing ratio data (3 data points) at 80-90 km and added them to improve the discussion. Furthermore, we revised the texts and figures based on all reviewer comments. We would like to emphasize that, nevertheless, these changes did not affect our conclusion.

We describe our point-to-point responses to each comment below with the original comments in black and our response in blue. The line number in the reply refers to the revised manuscript with a track change.

Abstract

Line 25: slow dust transportation compared to atmospheric expansion  The meaning of this sentence is unclear

We revised the text at line 28.

Line 28: oxygen decrease via downward transportation  the main text appears to indicate that there are different scenarios that could explain the decrease in the O abundance in the exosphere.

We revised the text in lines 29-30.

Main text

Lines 42-45: The ascent of water to high altitudes is not only driven by dust storms, but also occurs every year close to during southern spring and summer (close to perihelion), even in the absence of major dust storms. Water increase during perihelion/dust storms is shown by Belyaev et al. (2021), and Alday et al. (2021) showed the expected seasonal variations in the H₂O photolysis (i.e., H production), which are mainly driven by the Mars-Sun distance even in the absence of dust storms.

We revised the texts at lines 46-51 and added Belyaev et al. (2021) to the reference.

Lines 48-50: While H₂O is the main reservoir of H in the atmosphere, the O in H₂O just represents a very small part of the O reservoir in the atmosphere (95% CO₂). Are the variations in the O upper atmosphere during dust storms caused by the H₂O increase at high altitudes, or by the expansion of the

whole atmosphere due to the intensified warming?

We revised the texts at lines 56-57. As discussed in lines 190-209, several possibilities were raised to explain the O variations, such as H₂O ice absorption, odd hydrogen catalysis, or downward O transportation.

Line 68: rapidly  rephrase with a more specific term. For example, something like: on a matter of X days

We revised the texts at line 79.

Lines 70-74: From those plots it indeed looks that the cloud base was not situated at the 10-20 km level, which does not experience any change because of the dust storm. It looks as if the cloud layer was in the 20-40 km range, and the intensified warming because of the storm raised this level to the 60-80 km level. The increase in H₂O at 40-60 km is very sharp, but also brief and transient, the increase in the 60-80 km level appears steadier. I think the interpretation is that the intensified warming in the middle atmosphere prevents the clouds to form at these lower levels, and water ice clouds are formed instead at much higher altitudes. I would consider including the 60-80 km level in Figure 1 to help this discussion.

We added the data of 60-80 km in Figure 1.

Line 84: demonstrate  show the time series

Fixed.

Line 87: as rapidly, similar to its behavior at lower altitudes  As rapidly as at the lower altitudes?

Fixed.

Lines 89-94: What about dust acting as cloud condensation nuclei? The water will condense on the dust grains and therefore as the water ice opacity increases the dust opacity will decrease. Therefore, I am not sure that the interpretation of strong advection up to 60 km and then a decline above this altitude is entirely correct. The lag in the dust opacity might just indicate that water ice clouds are being formed at those altitudes.

The reviewer's opinion might be true, and we revised the text in lines 113-118.

Line 97: There is just one data point that exceeds 100 ppm at 40-60 km. In addition, the coverage of the SPICAM H₂O profiles is quite sparse. It should be addressed in the text how representative the SPICAM profiles are for representing the whole atmosphere.

It was a mistake. We meant 60-90 km and revised the text at line 112. (Please note that we added the 80-90 km data.)

Regarding the SPICAM data coverage, the SPICAM observations did not cover whole latitude and longitude (see Supplementary Fig. 2). Nevertheless, the increasing trend of H₂O mixing ratio is similar between northern and southern hemispheres in 20-40 km and 40-60 km altitude ranges. In the 60-80 km and 80-90 km ranges, we only cover the southern hemisphere, but the increasing tendency resembles those in the lower altitude. With that, we assume that SPICAM measurements are representative of the whole atmosphere. We added the explanation on lines 108-110.

Lines 98-100: This interpretation that the increase in H₂O in the middle atmosphere comes from a release of water by dust particles needs proof, and it might not be correct. The warmer temperatures due to the dust heating prevent the formation of water ice clouds, which essentially confine water vapour below the cloud level. In addition, the intensified atmospheric transport during dust storms

increase advection and raise water to the upper layers (e.g., Heavens et al. 2019, GRL; Shaposhnikov et al. 2019, GRL).

We removed this interpretation because there was no proof.

Line 106: its  this

Done

Lines 109-110: Is the Ly-beta emission optically thin? This will determine whether there is a linear relation between the density and the brightness.

Although it is not straightforward to estimate the optical depth from the disk (and limb) average data of Hisaki, the Ly- β emission is optically thick based on brightness value and the line-center scattering cross section (Masunaga et al., 2020). However, with the Chamberlain model and exospheric parameters close to our observation period in Chaffin et al. (2018), it turned out that Ly- β optical depth at exobase altitude (200 km) is near unity (1.2) on the disk but much larger than that on the limb. Thus, we can assume that our Ly- β brightness variation would change proportionally to variations in brightness on the disk rather than limb. So, we can assume that although the Ly- β brightness does not exhibit a linear relation to the column density near the limb, our observation most likely reflect the column density above 200 km altitude on the disk. We revised the text at lines 135-138.

Line 120: was much larger  provide some estimates of the % of variation.

The variation level is 20-50% as revealed in the residual analysis later in the text. Because we have not discussed the variation level at this point, we decided not to add the value here.

Line 122: Are the oxygen emissions optically thin? See comment above.

The 1304 emission is optically thick (Ritter et al., 2019) and its optical depth is much larger than Ly- β (Masunaga et al., 2020). On the other hand, 1356 is optically thin (Ritter et al., 2019). Thus, we used 1356 to discuss O column density variations in this study.

Line 125: in accordance with the observed atmospheric expansion  is the increase or the decrease of O in accordance with the atmospheric expansion?

We removed this text.

Lines 125-126: The decrease in the O densities during dust storms has been reported before and should be acknowledged in the text.

We added a discussion at 186-187.

Lines 128: Does turbulence only affect the O transport, or all gaseous species?

Eddy diffusion transports minor components downward with respect to the main CO2.

Line 137: 130 km is the peak intensity  this should be mentioned before making the calculation of the downward oxygen transportation.

We removed the discussion that calculated transportation speeds because our interpretation was changed after analyzing NGIMS data as another reviewer asked. We decided to only discuss the time scales of H and O variations in this study.

Lines 126-146: It seems that several scenarios are suggested that could explain the decrease of O in the upper atmosphere. I suggest to re-organise this long paragraph to clearly highlight how these different scenarios would work.

We reconsidered the discussion and reorganized this paragraph at lines 190-209.

Lines 148-149: The solar flux in Figure 2a also appears to oscillate. Is the frequency of the solar flux variations similar to the variations in the H or O brightness? That would not explain the anti-correlation between the O and H oscillations, though.

We agree that solar flux also shows some oscillations, but their variation is just ~5% as written in line 125. Because the hydrogen and oxygen airglow, or column densities, vary by at least 20%, we expect that their variations are caused by the change in their abundance, rather than the change in g-factor controlled by the solar flux.

Line 150: Are these variations also observed in the 1304 O brightness?

We did not see exactly the same variations in OI 1304, because OI 1356 is only excited by photoelectron impact in the ionosphere but OI 1304 emission is excited by both photoelectron impact and resonant scattering. Furthermore, OI 1304 is optically thick and it is thus not straightforward to discuss the variations of the column density with this emission line.

Lines 151-152: Here the analysis focuses on the dust storm. Are these oscillations also present in non-dusty periods?

Because we, unfortunately, have not observed Mars continuously for a long period (i.e., for a month) in the non-dusty period so far (Masunaga et al., 2020), we would like to leave the answer for our future study. We actually plan to make new observations in non-dusty periods by the Hisaki space telescope in late 2022 to early 2023 to search for the feature of the airglow variations and the difference from dusty periods.

Lines 164-169: As mentioned in the text, the surface measurements are representative of a very specific location of the surface, while the airglow measurements encircle the whole planet, making their relation

not that straightforward. Are similar oscillating patterns like these observed in the MCS temperature data at different altitude levels? The MCS dataset is likely more representative of the global behaviour of the atmosphere and will indicate how these waves propagate.

In our observation period, we used data only between 0-80 km altitudes because there were mostly no data available above 80 km in LT 11-13h (i.e., not enough data was successfully retrieved). Thus, it is difficult to discuss the wave properties at higher altitudes.

Line 171: Is the downward O transportation the reason of the decrease in the O brightness? From the paragraph before it seemed there could be several explanations.

Downward O transportation is one possibility to cause O depletion. Other possibilities are the uptake of O atoms by water ice and the acceleration of oxygen loss via odd hydrogen catalysis. We revised the texts on lines 190-209.

Line 178-189: Reference 4 (Chaffin et al. 2021) is for a regional dust storm in MY34. Stone et al. (2021) (reference 5) also indicate variations in the water abundance at high altitudes in a time scale of about 2 days for the global dust storm in MY34.

We removed this discussion due to changes in our interpretation but added a new discussion about the time scales of hydrogen atmosphere and high-altitude water at lines 138-161.

Lines 181-184: This should be rephrased. While the transport timescale might be like that, H atoms formed at 20 km will mostly not make it to 130 km, since the timescale for photochemical H loss in the lower atmosphere is much shorter (Gonzalez-Galindo et al. 2005, JGR).

We removed this discussion due to changes in our interpretation.

Figure 1

In my opinion, it would be more intuitive to put the panels in the reversed order: 0-20 km and the highest altitude range the uppermost panel, as if they were all sharing the same y-axis.

We have fixed the figure as the reviewer suggested. We also added the 60-80 km data.

Do the plotted lines include errorbars? They are averages, but they could include the standard deviation of all the averaged points to give an idea of the planet-wide variability.

As seen from Supplementary Figs. 1 and 7, the point-by-point data (grey colored dots) show large variations, resulting in large standard deviations. We tried to include the standard deviation in Figures 1 (and 2), but the wide y-range makes it difficult for readers to see the average variations that we want to discuss in this paper.

Additionally, because each average data point is an average value calculated by data points within LT 11-13h of a Martian day, a large standard deviation means that the data highly varies within a Martian day, suggesting that wave activities shorter than a Martian day exist. Because we focus on longer time scale (i.e., 6-8 day) variations in this paper, such a short time scale variation is out of our scope, and conclude that this does not affect our conclusion.

Therefore, we decided to show the average variations without standard deviations in Figures 1 (and 2).

I think this figure should also include the 60-80 km range for the plotted parameters to easily follow the discussion and interpretation of the climatological parameters.

We added the 60-80 km range in Figure 1 and changed the order.

Figure 2

The flux at 305A is not mentioned at all in the text.

We added an explanation at line 165.

As mentioned, some of the panels for the 60-80 km range could be included instead in Figure 1.

Done.

It must be indicated in the caption what the arrows represent.

We removed these arrows due to changes in our interpretation.

Figure 3

Include errorbars in the Residual brightness.

Done.

Include the 1304 A emission and search for similar patterns in MCS data.

Because the OI 1304 emission is optically very thick ($\tau > 30$), it is inappropriate to discuss column density variation. Thus, we decided not to include the 1304 variation here.

Methods

Lines 333-340: I guess that this dataset relies on the limb-viewing observations. Indicate the number of observations and coverage used in this study.

We added the geometry information of MCS limb observations in 20-40 km altitude in Supplementary Data Fig. 7. As seen from the figure, the limb observations were made at high latitudes. Because each limb observation had the same time stamp in the dataset, data in other altitudes have the same geometry information.

Lines 350-351: This should be included in the caption of Figure 2.

Done.

Line 355: has collected a comprehensive

Fixed.

Line 353: For SPICAM please also indicate the coverage and number of observations.

We added the SPICAM's geometry information in Supplementary Fig. 2. Unfortunately, the number of observations was not provided in the publicly opened dataset of Fedorova et al. (2021) (<https://data.mendeley.com/datasets/vx4gks6bx7/1>).

Again, we would like to thank the reviewer for reading our manuscript carefully and giving helpful comments to make this paper's quality better. We hope that the revised manuscript is suitable for publication in Nature Communications.

Sincerely,

Kei Masunaga

REVIEWERS' COMMENTS

Reviewer #1 (Remarks to the Author):

Alternate oscillations of Martian hydrogen and oxygen upper atmospheres during a major dust storm

Masunaga et al. re-submitted after revision to Nature Communications

This paper reports observations of the upper atmosphere of Mars with the Hisaki mission in Earth orbit that show the change in densities of H and O during the 2016 dust storm. Data from Mars Climate Sounder and MAVEN are now shown to track the changes in the atmosphere and progression of the dust storm. The conclusions of the work have been revised, and the authors now conclude that the H abundance increased while the O abundance temporarily decreased, and that there were alternate oscillations in the densities of H and O consistent with atmospheric waves. They conclude that the greater increase in H escape will lead to a change in oxidation state of the overall atmosphere, again likely tied to waves related to the coupling of the lower and upper atmospheres.

Overall the authors have been highly responsive to the comments from this referee on the original submitted manuscript. They have pulled up the MAVEN data from the same time period to compare global and local measurements, and apologies for confusing the dust storms in 2016 and 2019 in my original comments. As the authors point out the conclusions of the paper are not affected by the year of the dust storm, although the 2016 event has been much less reported upon which increases the significance of the present work.

At this point I think that the authors have done their job in revising the paper and defending their conclusions. It is an interesting and important result of interest to a broad range of researchers, and worthy of acceptance in Nat. Comm.

Having said that, here are a few specific points for the authors to consider before the paper is published:

- I remain somewhat skeptical about the significance of the relatively noisy O data, but it is clear that the H is increasing faster than the O. That is what counts in reaching the author's conclusions.

- In contrast to the statement in lines 148-150 that the O 1356 emission reflects just the density of O, it depends on both the O density and the electron collision rate. The authors may argue that the electron rate is relatively constant, but this should be stated. The text goes on to discuss this. I recommend stating that the O 1304 is more directly indicative of the O density, since it is dominated by resonant scattering, although there is still the question of increasing temp. that would broaden the line and increase the emission.

- In terms of historical measurements of the changing H density and escape rates, there have been several papers published by D. Bhattacharyya on the HST data and at least one of these should be referenced along with other works. I recommend referencing this summary paper of the HST observations:

"Seasonal Changes in Hydrogen Escape from Mars through Analysis of HST Observations of the Martian Exosphere near Perihelion", D. Bhattacharyya et al., J. Geophys. Res., 122, doi:10.1002/2017JA024572 (2017)

- It would be good to state somewhere why Ly-beta was reported rather than the much brighter Ly-alpha. I am guessing it is because of the high background of geocoronal emission from low earth orbit, but it should be explained.

- It is stated that since Ly-beta roughly doubled the density likely doubled. It is more likely that while the density increased, the higher temp. in the upper atmosphere broadened the H line

reflecting more solar emission, and the measured increase in brightness was due to a combination of these effects.

- For the record the 1304 emission is "mainly" produced by resonant scattering although there is a component from electron collisional excitation. This is supported by MAVEN observations of the 1304 triplet line ratio, which is consistent with resonant scattering and not electron collisions. As the authors agree the 1356 emission is excited by electron collisions, so the 1304 and 1356 emissions reveal different processes. Nonetheless they seem to track each other in the Hisaki data, at least well enough to not quibble about the difference.

RESPONSE TO REVIEWERS' COMMENTS

Reviewer #1 (Remarks to the Author):

Alternate oscillations of Martian hydrogen and oxygen upper atmospheres during a major dust storm

Masunaga et al. re-submitted after revision to Nature Communications

This paper reports observations of the upper atmosphere of Mars with the Hisaki mission in Earth orbit that show the change in densities of H and O during the 2016 dust storm. Data from Mars Climate Sounder and MAVEN are now shown to track the changes in the atmosphere and progression of the dust storm. The conclusions of the work have been revised, and the authors now conclude that the H abundance increased while the O abundance temporarily decreased, and that there were alternate oscillations in the densities of H and O consistent with atmospheric waves. They conclude that the greater increase in H escape will lead to a change in oxidation state of the overall atmosphere, again likely tied to waves related to the coupling of the lower and upper atmospheres.

Overall the authors have been highly responsive to the comments from this referee on the original submitted manuscript. They have pulled up the MAVEN data from the same time period to compare global and local measurements, and apologies for confusing the dust storms in 2016 and 2019 in my original comments. As the authors point out the conclusions of the paper are not affected by the year of the dust storm, although the 2016 event has been much less reported upon which increases the significance of the present work.

At this point I think that the authors have done their job in revising the paper and defending their conclusions. It is an interesting and important result of interest to a broad range of researchers, and worthy of acceptance in Nat. Comm.

We would like to thank again the reviewer for reading the revised paper and providing comments to improve our manuscript. We have considered all the suggested comments and revised our manuscript. We describe our point-to-point responses to each comment below with the original comments in black and our response in blue. The line number in the reply refers to the revised manuscript with a track change.

Having said that, here are a few specific points for the authors to consider before the paper is published:

- I remain somewhat skeptical about the significance of the relatively noisy O data, but it is clear that the H is increasing faster than the O. That is what counts in reaching the author's conclusions.

We believe that the O variations (the temporal decrease and periodic variations) are a real feature because the two independent measurements (insitu-NGIMS and global-Hisaki) exhibited reasonably similar variations. Regarding the H and O variations, the conclusion (more H escape and less efficient O escape) does not change even if only H increases in the upper atmosphere. Therefore, we would like to keep our discussion and conclusion as it is.

- In contrast to the statement in lines 148-150 that the O 1356 emission reflects just the density of O, it depends on both the O density and the electron collision rate. The authors may argue that the electron rate is relatively constant, but this should be stated. The text goes on to discuss this. I recommend stating that the O 1304 is more directly indicative of the O density, since it is dominated by resonant

scattering, although there is still the question of increasing temp. that would broaden the line and increase the emission.

We agree with the reviewer that the OI 1356 depends on both O density and photoelectron collision rates. Because there were no photoelectron observations at 130 km altitude in this time period, we instead used the solar 305 Å flux that causes photoionization of thermospheric components such as CO₂. Because the solar 305 Å flux was relatively stable (~5% variations as seen in Fig. 2a) compared to the OI 1356 airglow variations (20-50%), we argued that photoelectron fluxes in the ionosphere were also stable. Because our explanation might have been unclear, we revised the texts in lines 167-173.

Regarding the OI 1304 emission, it is correct that the OI 1304 emission is mainly excited by resonant scattering, but we have to take into account that OI 1304 is an optically thick emission at Mars. This means that its brightness is not linearly related to O abundance and that radiative transfer modeling to account for multiple scattering is required to link this brightness to O abundance. In this paper, therefore, it is demonstrated that OI 1356, which is an optically thin emission, is more directly indicative of O column density than OI 1304. We revised the texts in lines 173-177.

- In terms of historical measurements of the changing H density and escape rates, there have been several papers published by D. Bhattacharyya on the HST data and at least one of these should be referenced along with other works. I recommend referencing this summary paper of the HST observations:

Seasonal Changes in Hydrogen Escape from Mars through Analysis of HST Observations of the Martian Exosphere near Perihelion; D. Bhattacharyya et al., J. Geophys. Res., 122, doi:10.1002/2017JA024572 (2017)

We added the above paper to the reference list.

- It would be good to state somewhere why Ly-beta was reported rather than the much brighter Ly-alpha. I am guessing it is because of the high background of geocoronal emission from low earth orbit, but it should be explained.

As mentioned in Masunaga et al. (2020), the detector's sensitivity near the wavelength of Ly-alpha significantly degraded in the middle of the mission, whose possible cause was recently identified as the constant exposure to the extremely strong Ly-alpha emissions of geocorona and planetary atmospheres. So, Ly-alpha data is currently available only at the beginning of the mission (Kuwabara et al., 2017, JGR space phys., doi:10.1002/2016JA023247). The detector near Ly-beta, on the other hand, did not experience such a degradation because Ly-beta emission was much weaker than Ly-alpha. In this paper, therefore, we used the Ly-beta data instead of Lyman-alpha. We added a brief explanation about the non-use of Ly-alpha data in the Method section (at lines 295-300).

- It is stated that since Ly-beta roughly doubled the density likely doubled. It is more likely that while the density increased, the higher temp. in the upper atmosphere broadened the H line reflecting more solar emission, and the measured increase in brightness was due to a combination of these effects.

First, we meant that the "column density" doubled, so we revised the texts at line 140.

Second, if we understood this comment correctly, the reviewer's concern was if the exospheric temperature affected the Ly-beta brightness. We are actually not able to constrain whether density or temperature affected the column density directly from the disk-average measurements of Hisaki, but we can roughly estimate it. The airglow brightness can be roughly estimated by $B = gN/1e6$ (B is brightness in Rayleigh, g is g -factor in s^{-1} , and N is column density in cm^{-2}). We know that the Martian exobase temperature is about 200 K with variations of a few 10s % over Mars Year, including dust storm seasons, and it is much smaller than the solar atmospheric hydrogen temperature (an order of $1e4$ - $1e6$ K depending on regions). This indicates that the width of the solar Ly-beta line is much larger than that of the Doppler width of the scattering cross-section of Mars's hydrogen exosphere. Thus, we can say that the exospheric hydrogen atoms resonant with the line center solar flux and that g is not sensitive to the Martian exospheric temperature.

We can also discuss whether the column density N is sensitive to exospheric temperature. By using the Chamberlain exospheric density model and changing the exobase temperature from 180 K to 210 K (180 K is a likely exobase temperature at the beginning of our observation period (see, MAVEN orbit 3716 of Chaffin et al., 2018) + 30 K heating; the 30 K heating at the exobase was assumed from the atmospheric heating observed in the lower atmosphere in Fig 1g.) with a fixed H density, it turned out that the H column density just increased by $\sim 10\%$. Thus, the effect of exospheric temperature was small compared to the density in this time period. The density increase of H was likely caused by dissociation of high-altitude water and subsequent H diffusion from 60 km altitude and above, and this discussion is already provided in lines 144-165.

- For the record the 1304 emission is "mainly" produced by resonant scattering although there is a component from electron collisional excitation. This is supported by MAVEN observations of the 1304 triplet line ratio, which is consistent with resonant scattering and not electron collisions. As the authors agree the 1356 emission is excited by electron collisions, so the 1304 and 1356 emissions reveal different processes. Nonetheless they seem to track each other in the Hisaki data, at least well enough to not quibble about the difference.

Thank you for pointing this out. We revised the texts and mentioned that OI 1304 is mainly excited by resonant scattering at lines 173-174.

Regarding the OI 1356 and OI 1304 variations, the variation pattern of the two emissions was similar to each other, likely because they reflect mainly O abundance variations under the relatively small variations of the solar 1304 Å and 305 Å fluxes (Fig 2a). As discussed above, the variation of the optically thin OI 1356 is directly related to the O column density variations while the optically thick OI 1304 is not.